# Identification and Analysis of Micro-Exon Genes in the Rice Genome

**DOI:** 10.3390/ijms20112685

**Published:** 2019-05-31

**Authors:** Qi Song, Fang Lv, Muhammad Tahir ul Qamar, Feng Xing, Run Zhou, Huan Li, Ling-Ling Chen

**Affiliations:** 1National Key Laboratory of Crop Genetic Improvement, Huazhong Agricultural University, Wuhan 430070, China; aries981@webmail.hzau.edu.cn (Q.S.); m.tahirulqamar@webmail.hzau.edu.cn (M.T.u.Q.); xfengr@mail.hzau.edu.cn (F.X.); 18162856086@163.com (R.Z.); lihuan2729@163.com (H.L.); 2Hubei Key Laboratory of Agricultural Bioinformatics, College of Informatics, Huazhong Agricultural University, Wuhan 430070, China; lvfang2009@163.com

**Keywords:** micro-exons, constitutive splicing, alternative splicing, ancient genes, domain

## Abstract

Micro-exons are a kind of exons with lengths no more than 51 nucleotides. They are generally ignored in genome annotation due to the short length, whereas recent studies indicate that they have special splicing properties and important functions. Considering that there has been no genome-wide study of micro-exons in plants up to now, we screened and analyzed genes containing micro-exons in two *indica* rice varieties in this study. According to the annotation of Zhenshan 97 (ZS97) and Minghui 63 (MH63), ~23% of genes possess micro-exons. We then identified micro-exons from RNA-seq data and found that >65% micro-exons had been annotated and most of novel micro-exons were located in gene regions. About 60% micro-exons were constitutively spliced, and the others were alternatively spliced in different tissues. Besides, we observed that approximately 54% of genes harboring micro-exons tended to be ancient genes, and 13% were *Oryza* genus-specific. Micro-exon genes were highly conserved in *Oryza* genus with consistent domains. In particular, the predicted protein structures showed that alternative splicing of in-frame micro-exons led to a local structural recombination, which might affect some core structure of domains, and alternative splicing of frame-shifting micro-exons usually resulted in premature termination of translation by introducing a stop codon or missing functional domains. Overall, our study provided the genome-wide distribution, evolutionary conservation, and potential functions of micro-exons in rice.

## 1. Introduction

Micro-exons are a set of small exons with lengths no more than 51 nucleotides. Previous studies identified some micro-exon genes from mammals, insects, and plants [1,2,3]. Although micro-exon genes distribute widely in various species, systematic recognition of them has not been performed in a long period. In initial studies, the identification of micro-exons was not simple due to its mismatch until canonical and noncanonical splice sites were considered in alignment corrections [4]. After that, many alignment tools were improved to be more sensitive to micro-exons, and some pipelines were developed specially for micro-exons identification [1,5].

Alternative splicing is a process in which precursor mRNA adopts different splice sites and contributes to the expansion of sequence diversity and function [6,7]. In general, micro-exons whose lengths are not multiples of three nucleotides could cause frame-shifting in splicing, which may trigger nonsense-mediated decay [8]. Therefore, some alternative isoforms appear to be unstable or have no function. In addition, several studies have revealed that the level of micro-exon splicing is related to physiological disease in human; for example, mis-regulation of alternative splicing for neural micro-exons mediated by nSR100 is associated with autism [1,9]. Micro-exons are also good objects to investigate mechanism of splicing. Some studies have revealed there are sets of regulatory sequences in micro-exons and flanking introns, which are called exonic and intronic splicing enhancers (ESE and ISE). These motifs are recognized and bound by RNA-binding proteins (RBP) and contribute to splicing efficiency, consistent with the fact that most micro-exons are included in transcripts [5,10,11].

Micro-exons could function in various ways, including alternative splicing, such as degradation of the transcripts via nonsense-mediated decay (NMD), altering protein domain architecture, and introducing novel post-translational modification sites [12]. There have been several studies revealing that mis-regulation of splicing level could relate to several diseases in humans, such as autism spectrum disorders and schizophrenia [1,13], while few comprehensive studies have been performed in plants. However, we have observed that quite a few micro-exons were located in protein domains of some rice genes affecting foundational functions, such as *OsHAP3* and *RSR1*, which can regulate chloroplast biogenesis and starch biosynthesis, respectively [14,15]. Thus, it is crucial to determine detailed properties and roles of micro-exons in rice.

Nowadays, two important *indica* rice lines, ZS97 and MH63, have been assembled and annotated, providing a good basis for genome-wide micro-exon detection [16]. We extracted micro-exons from the genome annotations and compared micro-exons identified from alignments between RNA-seq data and genomes. Based on transcriptome data in different tissues and conditions, we aimed to explore different splicing types of micro-exons and the conservation of micro-exon genes in related species, and revealed the changes of protein structures caused by micro-exon splicing. We found that micro-exons accounted for 6.3% of all exons annotated by genomic information, and the lengths of 40.7% micro-exons were multiples of three nucleotides. In addition, more than half of micro-exon genes were evolutionarily ancient genes, and less than one-sixth were evolutionarily young genes, which only occurred in *Oryza* species. Gene ontology (GO) analysis showed that micro-exon genes varied in multiple metabolic and biosynthetic processes and enriched ribonucleotide binding. Further studies illustrated that micro-exons tended to be located in protein domains, such as protein kinase domain, RNA binding domains, etc. Alternative splicing of micro-exons could result in local structural recombination, while frame-shifting micro-exons might lead to premature termination of translation. Our study systematically analyzed the distribution, evolution, and functional information of micro-exons in rice genome.

## 2. Results

### 2.1. Identification and Inclusion Ratio of Micro-Exons in ZS97 and MH63

There were 16,508 and 17,517 micro-exons in the genome annotation of ZS97 and MH63, respectively, which accounted for ~6.3% of all the annotated exons. About 23% of the annotated genes contained micro-exons, implying that micro-exons were widely distributed in rice genomes. Among these annotated micro-exons, more than half (8816 in ZS97 and 9399 in MH63) were internal micro-exons, and the first or last micro-exons tended to be shorter than internal micro-exons (Figure 1A). Next, we employed RNA-seq data from four tissues (seedling shoot, root, flag leaf, and young panicle) of ZS97 and MH63 in four grown conditions (high/low temperature and long/short daytime) to identify micro-exons. In total, 7645 and 8137 internal micro-exons were identified in ZS97 and MH63, respectively. Compared with the above internal micro-exons identified in genome annotation, 2517 and 2824 were newly identified micro-exons in ZS97 and MH63, respectively (Figure 1B). Among these newly identified micro-exons, 1861 and 2079 were located in annotated gene regions, indicating that they might be un-annotated alternative splicing micro-exons. In total, the lengths of 40.7% micro-exons were multiples of three nucleotides, which were beneficial for the stability of open reading frames (ORFs) [1,5], whereas the others, which were not multiples of three nucleotides, could result in frame-shifting and encode different amino acid sequences.

### 2.2. Evolutionary Age of Genes Containing Micro-Exons

We investigated the evolutionary age of genes containing micro-exons by a previously reported approach [17,18]. The protein sequences of micro-exon genes were aligned to the non-redundant (NR) protein database in 13 taxonomic levels (details in the methods section). Then 5123 micro-exons of 3565 genes in MH63 were employed to construct their phylostratigraphic profiles (Figure 2A). We assigned a phylostratum (PS) value for each gene and defined the genes from PS1 to PS3 as “old genes”, while genes from PS11 to PS13 were defined as “young genes”. In total, 54.2% and 13.2% micro-exon genes were divided into old and young genes, respectively. After that, we compared the coding sequence lengths between young and old genes and observed that the average coding length of young genes was longer than that of old genes (Figure 2B, Wilcoxon rank-sum test, *p*-value < 0.001), but the lengths of their micro-exons were similar (Figure 2C). The average gene expression level of old micro-exon genes was much higher than that of young micro-exon genes in all the flag leaf, panicle, and shoot and root tissues (Figure 2D, Wilcoxon rank-sum test, *p*-value < 0.001), indicating that old micro-exon genes might perform more fundamental or essential functions than young micro-exon genes.

### 2.3. Conservation of Domains and Micro-Exons

In order to study the characteristics of micro-exons, we explored the conservation of micro-exons in angiosperm. McScanX [19] was employed to gain the collinear gene pairs among MH63 and ZS97, seven other representative *Oryza* species, *Zea mays*, and *Arabidopsis thaliana*. It was revealed that most micro-exons were highly conserved in *Oryza* species, owning the same open read frames in coding sequence and encoding the same amino acids. When we compared micro-exons in MH63 with those in *Zea mays*, there were 2509 micro-exons conserved in the open read frame and length, and about half of them had nucleotide substitutions, implying that micro-exons were relatively conserved in monocotyledons. However, when comparing MH63 with dicotyledon (*Arabidopsis thaliana*), only 197 micro-exons were conserved (Table 1), indicating that the micro-exon genes were highly divergent in monocotyledon and dicotyledon. Based on the above evolutionary age analyses, we speculated that micro-exon genes might originate from evolutionary old genes, and had independent differentiation and function in monocotyledon and dicotyledon.

To further investigate the protein functions related to micro-exons, we detected phosphorylation sites and domains in the micro-exon genes. Only 44 phosphorylation sites were identified in 30 micro-exons, indicating that only a few micro-exons may be involved in the function of cell signal transduction. On the other hand, about 58% micro-exons were located in domains. We then compared the enriched domains containing whole/part of micro-exons and their upstream/downstream domains, which did not include micro-exons. Table 2 showed the top ten domains containing micro-exons and the neighboring domains not containing micro-exons. Protein kinase domain (PF00069), RNA recognition motif (PF00076) and WD domain, and G-beta repeat (PF00400) were commonly enriched in micro-exon containing domains and their upstream/downstream domains. AP2 domain (PF00847), K-box region (PF01486), glycosyl hydrolase family 1 (PF00232), and protein tyrosine kinase (PF07714) were only enriched in domains containing micro-exons. The above results indicated that micro-exons might be related with functions of protein kinase and RNA recognition. Additionally, we found that the proportions of consistent domains (>95%) in gene pairs with conserved micro-exons in different species were higher than the ratios of non-conserved micro-exons (Figure 3A). Figure 3B–E showed four types of micro-exon related domains in collinear gene pairs between ZS97 and MH63. The first type was that micro-exons in collinear gene pairs had the same gene structure and were located in the same protein domain (Figure 3B); the second type was that the gene structures of collinear gene pairs were different, and only the micro-exon in one genome was located in the protein domain and the other genome was not (Figure 3C); and the third and fourth types were that the homolog sequences of micro-exons in one genome were contained in a longer exon in the other genomes, which formed similar or different protein domains (Figure 3D–E).

### 2.4. Quantification of Micro-Exon Usage and Gene Ontology Enrichment Analysis

To measure the constitutive or alternative splicing ratio of micro-exons, percent spliced-in (PSI) values in all samples were calculated by using the reads mapped to splicing junctions. In total, 4921 and 5162 micro-exons in ZS97 and MH63, respectively, had PSI values > 0.1 in at least one RNA-seq dataset. More than 70% of micro-exons tended to be constitutively spliced (CS; PSI ≥ 0.9) in all samples, which was consistent with the results in humans [5], and the other micro-exons were alternatively spliced (AS) in different tissues (Figure 4A). The PSI values in the same tissue (flag leaf/root/panicle/shoot) under different conditions were similar, suggesting that temperature and light had little effect on gene splicing.

Hundreds of micro-exons had various splicing types in different tissues, but only one micro-exon was AS across all samples in MH63. As seen in Figure 4B, 3679 micro-exons were expressed in the four tissues, and 838 micro-exons were expressed in two or three tissues. In addition, 41, 110, 47, and 296 micro-exons were specifically expressed in flag leaf, root, shoot, and panicle, respectively. Then we performed GO analysis of these tissue-specific micro-exon genes and observed that the enriched functions were highly divergent in different tissues (Figure 4C). The common functions in the four tissues mainly enriched in the ‘cellular aromatic compound metabolic process’ and ‘heterocycle metabolic process’. Furthermore, the ‘organic substance metabolic process’ was enriched in flag leaf and shoot, while the ‘cellular nitrogen compound metabolic/biosynthetic process’ and ‘aromatics compound/heterocycle/nucleobase-containing compound biosynthetic process’ were enriched in the panicle, root, and shoot. However, the ‘macromolecule metabolic process’, ‘protein metabolic process’, and ‘phosphorus metabolic process’ were only enriched in the flag leaf; the ‘reproductive process’, ‘RNA biosynthetic process’, and ‘regulation of RNA metabolic process’ were merely enriched in the panicle; the ‘regulation of (cellular) biosynthetic process’, ‘multicellular organismal process’, and ‘developmental process’ were only enriched in the root, indicating that these specifically expressed micro-exon genes played different roles in the four tissues.

### 2.5. Structure and Functional Analyses of Alternative Spliced Micro-Exons

To explore the influence of AS micro-exons in protein structure and function, we predicted and analyzed the three-dimensional (3D) structures of some representative micro-exon genes. Generally, AS micro-exons with a multiple of three nucleotides only changed the local sequence and structure, and the global protein structure changed little, whereas AS micro-exons with lengths that were not multiple of three could cause frame-shifting and great changes in amino acid sequences and 3D structures [20]. Figure 5A–D showed four examples of 3D structures containing AS micro-exons. As seen in Figure 5A, the MH05t0027800-01 protein (rice starch regulator 1, *RSR1*) contained two Apetala 2 (AP2) domains; one AP2 domain included a 45-nt micro-exon with three DNA binding sites [15,21], and another downstream AP2 domain had a similar 3D structure. Kim et al. reported that the AP2 domain had an 18 amino acid core region that formed an amphipathic α-helix, which was completely overlapped with the micro-exon [21]. It was interesting that the 45-nt micro-exon was CS in flag leave and shoot but AS in panicle and root. Considering that *RSR1* acts as a transcription factor in starch synthesis [21], this variation might have different effects on starch synthesis in different tissues. Figure 5B showed the structure of the MH01t0092800-26 protein (*OsGI*) with or without a micro-exon. When the micro-exon was not included, it only affected the local structure but did not change the overall structure of the protein. A previous study revealed that *OsGI* was expressed in the biological clock and had different effects on flowering time in long days or short days [22]. The micro-exon was AS in flag leaf and CS in root and shoot, indicating its dynamic effect in different tissues and conditions. Besides, the *OsGI* gene had another micro-exon and various isoforms. Figure 5C showed the structures of MH03t0653800-01 protein (*OsSUV3*) containing or not containing a 46-nt AS micro-exon, whose function is related to high salt resistance [23]. The exclusion of the micro-exon led to the conformation change of the *OsSUV3* protein C-terminal due to the 46-nt micro-exon located at the back end of the gene. Figure 5D illustrated the structure of the rice heterochronic gene MH07t0148200-01 (SUPERNUMERARY BRACT, *SNB*), whose conformation is much different from the structure not containing a 31-nt AS micro-exon. It was interesting that *SNB* genes also had two AP2 containing micro-exons. Previous studies reported that mutants of the rice *SNB* gene with an insertion in the terminal region could cause significant sterility [24]. Taken together, most micro-exons with lengths that were not multiples of three nucleotides could change the open reading frame and cause overall structure changes after missing micro-exons.

## 3. Discussion

Previous studies have reported some micro-exons and their roles in various species, but there is no systematic study of micro-exons in plants. In this study, we used RNA-seq data from various rice tissues to identify thousands of micro-exons. About one third of them were not annotated but located in the region of the annotated genes. While more than half of the micro-exon genes were ancient genes, the other micro-exons were conserved in *Oryza* species and other monocotyledons while not conserved in dicotyledons, implying that these genes might have an ancient origin but differ in later evolutionary processes. Moreover, most micro-exons were located in protein domains and generally tended to be CS. All these results suggested that the micro-exon genes may play crucial roles in primary functions of the organism, and stable splicing of micro-exons was essential for maintaining their functions.

The results of GO analysis indicated that the micro-exon genes were enriched in many biological processes, and they were enriched in ribonucleotide binding in molecular function. We found that a large number of domains containing micro-exons were related to DNA or RNA binding, such as AP2 and RNA recognition motif, which may be a key role of micro-exons in molecular functions. It has been reported that splicing mis-regulation of brain-expressed micro-exons led to brain-related diseases [1]; therefore, many micro-exon genes have important functions. Furthermore, the tissue-specific protein contained disordered regions and conserved binding motifs [25]. Although most micro-exons were widely expressed in the four tissues, genes containing tissue-specific micro-exons also had distinctive functions in ZS97 and MH63.

Previous studies reported three effects of micro-exons: leading to premature stop codons, changing the protein structure, and creating sites for post-translational modification [12]. Only a few phosphorylation sites were found in the micro-exons of ZS97 and MH63, implying that phosphorylation only represented a few cases of post-translational modification. On the other hand, most micro-exons were contained in domains, and some of them harbored functional sites. Besides, the secondary structure of micro-exons usually consisted of several types, including α-helix, β-strand, and coil. There were two AP2 domains in *RSR1* [21], and a 45-nt AS micro-exon lied in one of the domains, harboring three DNA binding sites. There is no doubt that micro-exon exclusion will affect gene function. As *RSR1* is a transcription factor, the splicing level of this micro-exon may affect the expression of downstream genes. Additionally, inclusion or exclusion of the micro-exon could easily cause premature terminal of translation, especially for the lengths that were not multiples of three nucleotides. In some cases, we observed many shortened sequences by alternative splicing, and the encoded protein functions can be influenced distinctly.

This study provided an overview on micro-exons in ZS97 and MH63, and demonstrated the relationship of protein structure, function, and micro-exon splicing. The splicing mis-regulation of micro-exons leads to disease in human [1,5]. In this case, a high proportion of micro-exons were CS, and they had precise and stable splicing process and conserved domains, providing a guarantee of regular metabolic process. In addition, the predicted domains and structures in this study illustrated that the changes among proteins were caused by AS micro-exon inclusion and exclusion to some extent. However, the predicted structures cannot completely explain the variation, and for many genes, their 3D structures were hard to predict. Despite the lack of software and structure data, the exact splicing of micro-exons in transcripts and proteins cannot be obtained only by RNA-seq data. Generally, the genes with micro-exons have various isoforms in the expression process. For instance, a 34-nt micro-exon located in *OsTrx1* is a histone H3K9 methyltransferase gene, but the micro-exon is not annotated in the annotation information. With advanced techniques, complete sequences and structures of the isoforms can be obtained through full-length transcript data and protein data, and then the precise roles of micro-exons in proteins revealed.

## 4. Materials and Methods

### 4.1. Materials and Data Preparation

To identify micro-exons in ZS97 and MH63, RNA-sequencing was performed from four tissues (flag leaf, panicles, seedling shoot, and roots) in four different conditions (high temperature and long day, high temperature and short day, low temperature and long day, and low temperature and short day. High temperature: 28–32 °C; low temperature: 22–25 °C; long day: 14 h light and 10 h dark; short day: 10 h light and 14 h dark). Besides, each sample had two biological replicates (Illumina HiSeq2000 platform). Finally, a total of 948 and 988 million strand-specific paired-end 101-nt reads were obtained from ZS97 and MH63 (an average of 29.6 and 30.9 million per sample), respectively. Genome data and annotation information of ZS97 and MH63 were downloaded from the website (http://rice.hzau.edu.cn; version RS1; accessed on 4 July 2016). Genome and annotation information of other plant species were retrieved from Ensembl Plants (http://plants.ensembl.org/; accessed on 12 April 2018).

### 4.2. Identification of Micro-Exons

Firstly, we extracted all micro-exons whose lengths were in the range of 3 to 51 nt from genome annotations. The micro-exon, as the first or last exon in genes, was examined as to whether it was coding sequences or untranslated regions. Then to identify the micro-exons in ZS97 and MH63 based on RNA-seq data, all the reads were mapped to their reference genomes using HISAT2 [26]. A strand-specific parameter was set in the alignments (--rna-strandness RF) and unique mapping reads with less than 2 mismatches were retained. For each mapping, we reserved at least 6-nt on both sides to completely align the micro-exons. In order to detect as many micro-exons as possible, the insertions supported by more than 10 reads in all samples were identified as candidate internal micro-exons. The candidate micro-exons were divided into annotated micro-exons and novel micro-exons according to the annotation. Compared with annotated genes, we detected the locations of the novel micro-exons. The distribution of length of novel and annotated micro-exons were measured and displayed, and the whole identification process is showed in Appendix A.

### 4.3. Evolutionary Age of Genes Containing Micro-Exons

Previous methods described that the protein sequences from the NR protein database were attributed to 13 taxonomic levels (PS1: Cellular organisms; PS2: Eukaryota; PS3: Viridiplantae; PS4: Streptophyta, Streptophytina; PS5: Embryophyta; PS6: Tracheophyta, Euphyllophyta; PS7: Spermatophyta; PS8: Magnoliophyta, Mesangiospermae; PS9: Liliopsida, Petrosaviidae, Commelinids, Poales; PS10: Poaceae; PS11: BOP clade; PS12: Oryzoideae, Oryzeae, Oryza; and PS13: *O. sativa*) [17]. The protein sequences containing micro-exons from MH63 were aligned to the 13 levels of non-redundant databases using BLASTP with *e*-values ≤ 1 × 10^–5^, identity ≥ 0.3, and coverage ≥ 0.8. The age of a gene was assigned the taxonomic level of its alignment, and the gene that failed to be aligned with all databases was categorized to PS13 (*O. sativa*). Then the micro-exons were divided as their corresponding types of genes. The genes of PS1–3 were assigned as “old genes”, while the genes of PS11–13 were “young genes”. Next, the length of CDS and micro-exons of the old and young genes were calculated and compared. Furthermore, the gene expression levels of the old and young genes in four tissues were displayed.

### 4.4. Conservation of Micro-Exons and Domains

The protein sequences of MH63 were aligned to the other species by BLASTP [27] with default parameters, and the collinear gene pairs were obtained using MCScanX [19]. Then the alignments of micro-exon regions were applied to measure the conservation of micro-exons. The micro-exons in the alignments, which had the same phase and length of CDS in both MH63 and another species, were considered as conserved ones. In addition, Interproscan [28] was used to predict domains in all the sequences. For each micro-exon, we compared the domains of micro-exons in MH63 with the ones in other species. At last, we also examined phosphorylation sites in the micro-exons via the Plant Protein Phosphorylation DataBase (P3DB, http://www.p3db.org/).

### 4.5. Percent Splice-In (PSI) Index of Micro-Exons

To measure the usages of micro-exons, the results of HISAT2 were used to establish a non-redundant transcript annotation with StringTie [29]. For each micro-exon gene, the longest transcript containing micro-exons was searched by the annotation information. Then we constructed transcripts that contained micro-exons and flanked 100-nt to measure PSI for each micro-exon. We also constructed a dataset without micro-exons but flanked 100-nt. The transcripts would be combined into one in the cases where the distance among multiple micro-exons were within 100-nt. After that, the RNA-seq data was mapped onto the constructed transcripts using BWA [30] (single-end; options mem) with no more than 2 mismatches. As shown in Appendix A, R_L_ and R_R_ represented the number of reads supporting the left and right junction for each micro-exon, respectively, while R_skipped_ was the number of reads spanning the junction where micro-exons were skipped. The counted reads should span the junction at least 3-nt. With these quantities, the number of reads supporting micro-exon inclusion, R_tot_, was computed as follows:R_tot_ = 2 min{R_L_, R_R_}(1)

Previous studies demonstrated that Equation (1) can avoid cases in which alternative 5′ or 3′ splice site biases the estimated micro-exon usage [5]. In addition, the sum of R_L_ and R_R_ (or R_skipped_) should be no less than 10, otherwise the PSI value should be considered missing. The PSI value was computed as follows:PSI = R_tot_ / (R_tot_ + R_skipped_)(2)

We combined the PSI values of two biological replicates by the following rules: if a micro-exon had PSI value only in one biological replicate, the value was used; otherwise, if both replicates had PSI values and their difference was less than 10%, we used the mean; other cases were assigned to missing.

Generally, micro-exons with PSI values in the range of 10–90% were categorized as AS micro-exons, and >90% of PSI were assigned as CS. For each tissue, micro-exons with PSI ≥ 10% in at least two samples were collected, and then the micro-exons and relevant genes were compared and displayed in four tissues.

### 4.6. Functional Annotation of Micro-Exon Genes

To detect the function of micro-exon genes, micro-exons with PSI ≥ 10% in at least two samples were collected and the genes containing tissue-specific micro-exons were retained. Then AgriGO [31] was performed to do GO enrichment analysis with the reference *Oryza sativa* subsp. *indica*. Furthermore, the genes containing AS micro-exons in each tissue were also analyzed.

### 4.7. Structures and Domains of Micro-Exons

To investigate the structure of micro-exons, the protein sequences of micro-exon genes were extracted from annotation information. The longest sequences were selected if micro-exon genes had several transcripts. Then the sequences with inclusion and exclusion of micro-exons were constructed. In addition, the unannotated sequences from transcript assembly and prediction were used as Appendix A. After that, the protein structure of some genes were predicted using RaptorX [32], and the domains were predicted with Interproscan [28]. Chimera [33] was employed to display and compare the protein structures including and excluding micro-exons.

## Figures and Tables

**Figure 1 ijms-20-02685-f001:**
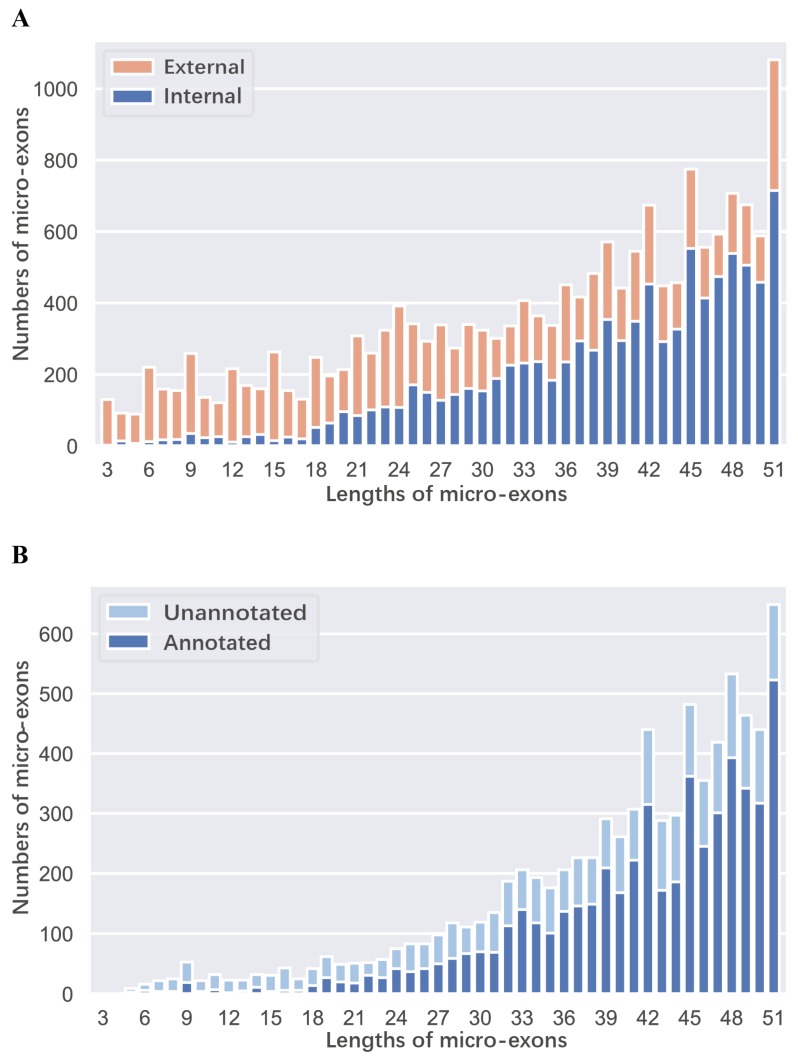
The length distribution of micro-exons in MH63. (**A**) The length distribution of external and internal micro-exons based on MH63 genome annotation. (**B**) The distribution of lengths for the annotated and unannotated micro-exons, which were identified from RNA-seq data in four tissues.

**Figure 2 ijms-20-02685-f002:**
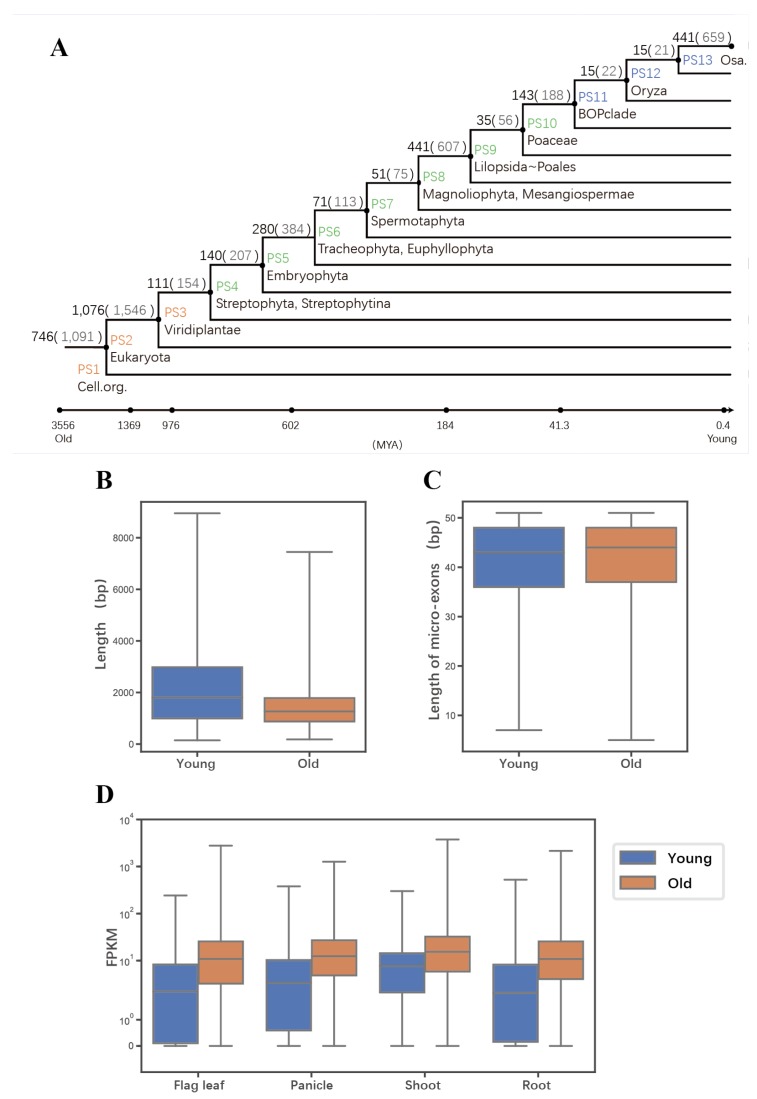
The evolutionary ages of micro-exons genes. (**A**) The phylogenetic tree shows the genes containing micro-exons in different evolutionary times, from phylostratum (PS)1 (single-cell organisms) to PS13 (*O. sativa*). There are two numbers on each branch: the numbers of genes corresponding to each level are outside the brackets, while the numbers of identified micro-exons are in brackets. The genes of PS1–3 are assigned as “old genes” while the genes of PS11–13 are “young genes”. (**B**) The comparison of coding sequence (CDS) lengths between the young genes and old genes. (**C**) The comparison of micro-exon lengths between the young genes and old genes. (**D**) The comparison of gene expression in four tissues between the young genes and old genes. FPKM represents pair-end fragments per kilobase of exon model per million mapped fragments. The blue color represents young genes and the orange color represents old genes.

**Figure 3 ijms-20-02685-f003:**
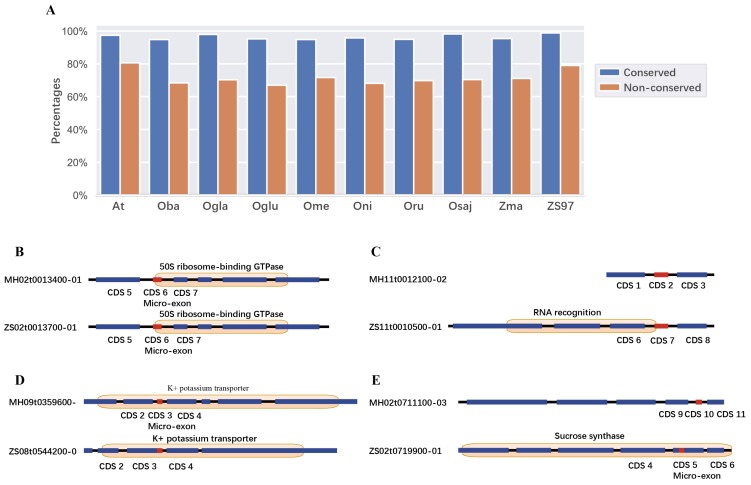
The proportion of conservation for micro-exons and domains in gene pairs. (**A**) The percentage of consistent domains between the conserved and non-conserved micro-exons. (**B**–**E**) Several examples of consistent and inconsistent domains with conserved and non-conserved micro-exons in ZS97 and MH63.

**Figure 4 ijms-20-02685-f004:**
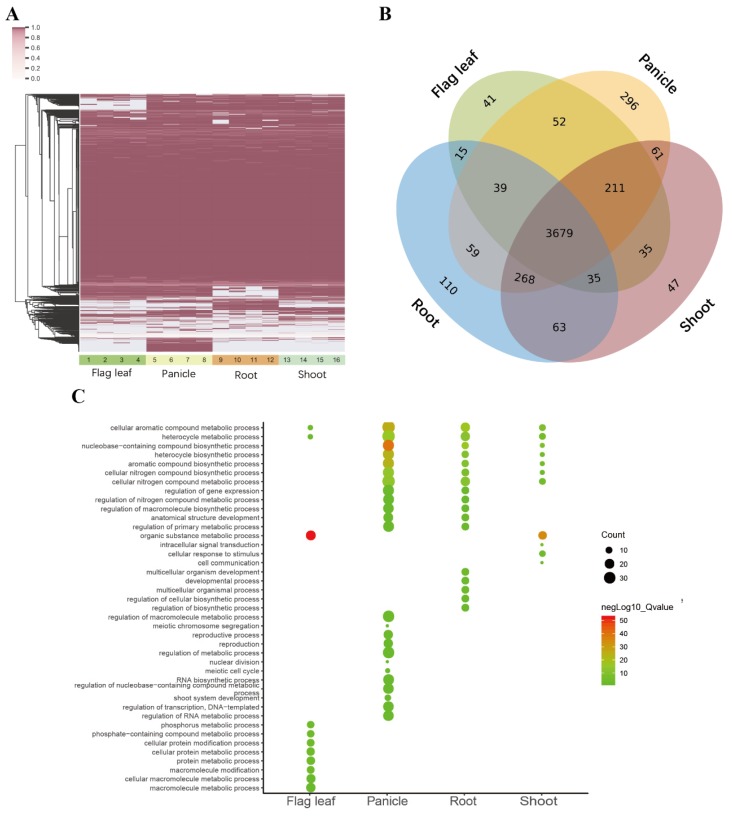
The percent spliced-in (PSI) values and gene ontology (GO) enrichment analyses in four MH63 tissues. (**A**) PSI of 5162 micro-exons in 16 samples. 1–4: flag leaf, 5–8: panicle, 9–12: root, 13–16: shoot in order of high temperature long daytime, high temperature short daytime, low temperature long daytime, and low temperature short daytime. (**B**) Micro-exons with PSI ≥10% in the four tissues. (**C**) GO analysis of tissue-specific micro-exon genes.

**Figure 5 ijms-20-02685-f005:**
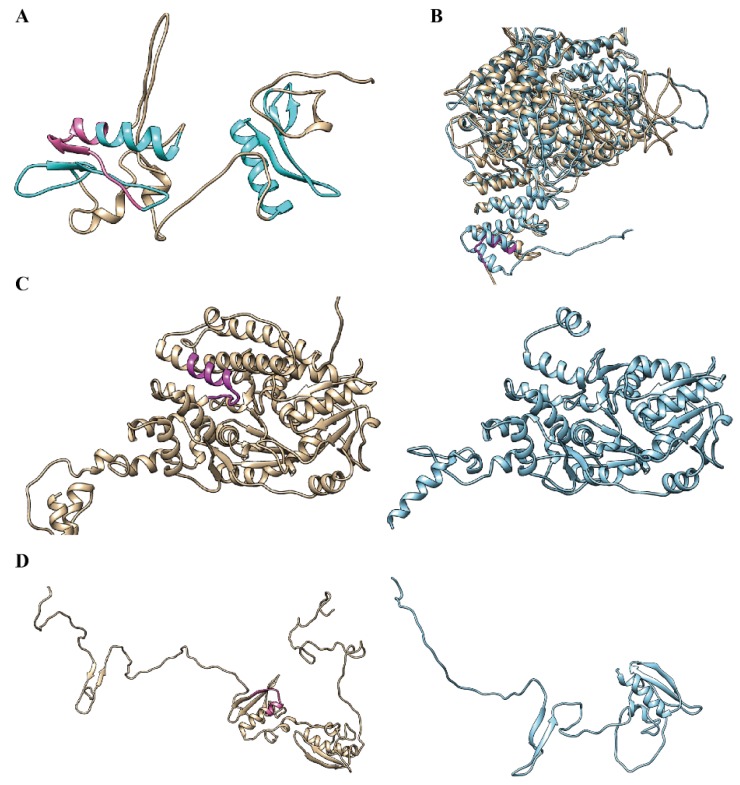
Structure comparison of proteins harboring micro-exons. (**A**) Structure of MH05t0027800-01 (*RSR1*). Two AP2 domains are presented in cyan and micro-exon is in pink. (**B**–**D**) Structural comparison of the proteins containing micro-exon (tan) or excluding micro-exon (cyan). (**B**) is MH01t0092800-26 (*OsGI*) with a 42-nt micro-exon, (**C**) is MH03t0653800-01 (*OsSUV3*) with a 46-nt micro-exon, and (**D**) is MH07t0148200-01 (*SNB*) with a 31-nt micro-exon. Micro-exons are highlighted in magenta.

**Table 1 ijms-20-02685-t001:** Conservation of genes containing micro-exons.

Species (Paired with MH63)	Total Gene Pairs	Gene Pairs with Micro-Exons	Gene Pairs with Conserved Micro-Exons
*Arabidopsis thaliana*	1828	253	214 (197)
*Oryza barthii*	27,306	3821	3183 (3470)
*Oryza glaberrima*	25,755	3559	2949 (3571)
*Oryza glumipatula*	27,314	3731	3111 (3434)
*Oryza meridionalis*	21,571	3156	2691 (2975)
*Oryza nivara*	27,192	3638	3089 (3422)
*Oryza rufipogon*	29,323	3864	3208 (3523)
*Oryza sativa*	26,237	3875	3217 (3561)
*Zea mays*	21,953	3524	2386 (2509)
ZS97	38,649	3747	3161 (4251)

**Table 2 ijms-20-02685-t002:** The top10 domains including and excluding micro-exons.

Rank	Domains Including Micro-Exons	Domains Excluding Micro-Exons
1	Protein kinase domain (PF00069, 66)	WD domain, G-beta repeat (PF00400, 163)
2	RNA recognition motif (PF00076, 42)	RNA recognition motif (PF00076, 78)
3	AP2 domain (PF00847, 36)	Protein kinase domain (PF00069, 60)
4	K-box region (PF01486, 28)	IQ calmodulin-binding motif (PF00612, 46)
5	Glycosyl hydrolase family 1 (PF00232, 24)	PPR repeat family (PF13041, 44)
6	Protein tyrosine kinase (PF07714, 23)	PPR repeat (PF01535, 39)
7	Myb-like DNA-binding domain (PF00249, 20)	SRF-type transcription factor (PF00319, 32)
8	Serine carboxypeptidase (PF00450, 20)	C2 domain (PF00168, 28)
9	WD domain, G-beta repeat (PF00400, 17)	Helicase conserved C-terminal domain (PF00271, 28)
10	Major facilitator superfamily (PF07690, 16)	Gelsolin repeat (PF00626, 25)

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
