# Peer review of "Identification and Analysis of Micro-Exon Genes in the Rice Genome"

_ijms, 2019, doi:10.3390/ijms20112685_

Round 1
Reviewer 1 Report
the bioinformatic analysis of the two rice genomes for micro-exons is timely and justified well. As mentioned micro-exons are not well know, so a few sentensis to clarify how they are identified would beusefull. E.g they are not identified by start codon followed by an open reading frame and terminated by a stop codon but rather the interaction between genomic sequences and RNAseq data. Based on this study rice genomes have approx 17,000 micro exon, with short genic sequences (3 to 51 nt or encoding 1 to 17 amino acid long peptides). Are here any bias in the codon usage? How is the distribution of the 20 amino acids, and compared with overall amino acid composition in rice? They have used RNAseq data from 2 replicates, their views on why that is sufficient as other recommed 3-4 replicates.
Author Response
We are grateful to reviewer for positive comments on our research article. Point to point answers to reviewer queries are enlisted in attachment.

Reviewer 2 Report
Authors investigated distribution, evolution, classification and splicing of Micro-exons that have no more than 51 nucleotides in length in rice. It is interesting topic. The source is useful. The concerns are the level of biological significance of the micro-exons and whether the results here represent a specific or shared pattern in plant or not. The true function of micro-exon is still unknown.
1. Introduction section: The significance of the micro-exon should be improved. The micro-exon is not new but why the authors want to do this while other researchers don’t do this micro-exon.
2. What is ratio of micro-exon was alternative spliced which will cause the reading frame change?
3. The micro-exon may be separated by micro-intron only. Does the micro-intron have same patterns as Micro-exon ?
4. The characteristic conclusion of micro-exon in rice is specific or representative of other plants?
5. The color in Figure 2D should be kept consistent with that in B and C
6. The method and introduction section: the citation should cite the original article. I found that some citations were wrong. E.g. reference 15. The authors cite this one for they used the method from ref 15. While ref 15 cited from another paper , then the other cites from others. Only the original papers that have the direct information should be cited. Do not misguide the readers and cite what you really read. Many cited articles in this manuscripts are wrong and need to be corrected.
Author Response

(The authors gave the same response as above.)

Round 2
Reviewer 2 Report
The authors revised the manuscripts with the recommendation from reviewers, and improved the manuscript. I recommend to get the manuscript accepted for publication after correcting any spelling errors.